# How Can Autonomy Support from a Coach, Basic Psychological Needs, and the Psychological Climate Explain Ego and Task Involvement?

**DOI:** 10.3390/ijerph20216977

**Published:** 2023-10-26

**Authors:** Arne Martin Jakobsen

**Affiliations:** Faculty of Education and Arts, Nord University, 8049 Bodø, Norway; arne.m.jakobsen@nord.no

**Keywords:** autonomy support, self-determination theory, psychological climate, ego involvement, task involvement

## Abstract

The aim of this study was to consider the relationships among the autonomy support an athlete perceives from their coach, the three basic psychological needs (autonomy, competence, and relatedness), the psychological motivational climate of the team (a task or ego climate), and the athlete’s motivational orientation (ego or task involvement). No other studies have investigated this. My three hypotheses were as follows: autonomy support from a coach will have an impact on motivational involvement, all three basic needs will have an impact on motivational involvement, and motivational involvement will be explained by the motivational climate. A total of 175 elite male ice hockey players from Norway, ranging in age from 15 to 18 years old, answered questionnaires about autonomy support, perceived motivational climate, achievement goal orientation, and basic psychological needs. A multiple regression analysis was conducted to predict ego–task involvement using autonomy support from the coach, the need for autonomy, the need for competence, the need for relatedness, the task climate, and the ego climate. The only two variables that statistically significantly predicted ego–task involvement were the autonomy support from the coach (std. beta = 0.28, sign = 0.05) and the ego climate (std. beta = 0.34, sign = 0.01). The analysis revealed that the athletes had a higher score on task (M = 4.85) than ego (M = 3.34) involvement, but when these were transformed into two variables (high and low) for task and ego involvement, we found that most players scored high for both task and ego involvement. We found that autonomy support from the coach had a positive relationship with a high score for players on both task and ego involvement. We also found that the three basic psychological needs had no impact on the motivational involvement of the athletes. Lastly, we found that the ego climate had an impact on motivational involvement. There was a positive relationship between a high score for the ego climate and a high score for both ego and task involvement.

## 1. Introduction

The self-determination theory (SDT) is a meta-theory of the autonomous control of behavior and voluntary engagement [1]. The SDT argues that motivation emerges from the interaction between social contextual factors and human nature [2]. The theory explains human behavior and motivation based on differences in motivational orientations, interpersonal perceptions, and contextual influences on motivation [3].

The SDT assumes that people are growth-oriented and actively seek optimal challenges. They will take part in activities for enjoyment and interest in the activity itself [1]. Furthermore, the theory assumes that maximal well-being and optimal performance occur when an athlete’s innate needs for autonomy, relatedness, and competence are satisfied [4]. Autonomy-supportive environments that satisfy these three basic needs tend to increase an athlete’s persistence, effort, and well-being [3,5,6,7,8]. A coach’s support for autonomy shows a strong positive correlation with intrinsic motivation, as well as integrated and identified regulation [9,10]. Athletes who perceive autonomy support from their coach have higher chances of scoring highly for autonomic regulation, according to the self-determination theory [11]. There is also a strong positive correlation between autonomy support from a coach and an athlete’s need for autonomy and relatedness. The association with the need for competence is in the upper-moderate range [9].

### 1.1. Theory of Basic Psychological Needs in Sport

One out of six mini-theories within the self-determination theory (SDT) is the basic psychological needs theory (BPNT) [1,6,12]. This theory elaborates on the concept of psychological needs and their relationships to psychological well-being [13]. According to the theory, psychological well-being and optimal functioning are predicated by autonomy, competence, and relatedness. Autonomy is based upon the inner endorsement of behavior. The perceptions of an individual’s behaviors are derived from themselves [9]. Relatedness requires a sense of mastery and succeeding at challenging tasks and a feeling of being cared for by significant others [9]. All three needs are essential, and if any are missing, there will be distinct functional costs [14,15]. The construct of basic psychological needs is the main predictor of human motivation [16].

### 1.2. Achievement Goal Theory

Task and ego involvement are the two main types of goals related to the achievement goal theory (AGT) [13,17]. The theory postulates a dichotomous model with two types of motivation: task and ego motivation. The AGT proposes that an athlete’s goal orientation and their perceived motivational climate will influence their behavior [16]. Task-motivated athletes are mainly interested in subjective indexes of success, such as tactical development, technical self-improvement, skill-learning, and the mastery of challenges [18]. Ego-motivated athletes look at themselves as successful only if they perform better than others, regardless of personal improvements in performance that they have achieved [18]. Beating others is the main preoccupation of ego-orientated athletes [17]. An individual’s performance in achievement-related activities is directed towards ego- or task-related goals [19]. A person can be more or less ego- and task-involved at different times during their task engagement. One theory, according to Bruner et al. [19], is based upon the idea that variations in people’s definitions of successful accomplishments and the judgment of their abilities are vital for understanding athletes’ motivational processes. In sports, the motivational climate perceived by an athlete influences the process of setting and pursuing goals. This is regardless of whether the athlete can be said to be task- or ego-oriented. Individuals’ behaviors and outcomes can be significantly influenced by their orientation, whether it is high-task/high-ego, low-task/low-ego, low-task/high-ego, or high-task/low-ego, during their participation in sport [18]. An interaction between an individual’s intrapersonal level and the motivational climate generated by social agents, such as, coaches and peers, determines their achievement of goals.

The intervening variable in the motivational climate is one important situational factor for the achievement of goals. This regulates the relationship between goal orientation and athletic performance [18,20,21]. The motivational climate is the perceived structure of the achievement environment, which is mediated by the coach’s attitudes and behavior [18,20,21]. We can identify two types of climates. When the athlete perceives that the coach compares and sets them up against other athletes and their mistakes are criticized and punished, we call it an “ego-oriented” climate [18,20,21]. On the other hand, in an “mastery” climate, the coach places an emphasis on personal effort and skill development [18,20,22]. There is a strong correlation between the sports climate and the autonomy support provided by the coach [20,22]. There is also support for the idea that an ego-involving climate undermines the need for competence, autonomy, and relatedness, whereas a task-involving motivational climate increases the perception of these needs [23]. A task environment is derived from the perceived support from the coach and task-oriented athletes [11].

A task-oriented climate has a perceived advantage over an ego-oriented climate. One possible reason for this could be that the athlete is encouraged to focus on factors within their control in a task-oriented environment. In an ego-oriented environment, athletes tend to use social comparison processes when assessing their own competence [12]. A task-oriented climate can be cultivated via the coach’s feedback, focusing on the athlete’s performance relative to self-referenced criteria of improvement and achievement [16].

Perceived ego-oriented climates are associated with negative worries and effects [24]. In contrast, perceived task-oriented climates are positively associated with outcomes such as intrinsic motivation, positive effects, perceived competence, and feelings of autonomy and flow [24].

The aim of this study was to consider the relationships among the autonomy support an athlete perceives from their coach, the three basic psychological needs (autonomy, competence, and relatedness), the psychological motivational climate of the team (task or ego climate), and the athlete’s motivational orientation (ego or task involvement). We computed the motivational involvement into groups of low ego and low task, low task and high ego, high task and low ego, and finally, high task and high ego.

In the study, we considered the basic psychological needs theory (BPNT) developed by Deci and Ryan [5,6,14] and aimed to determine how well the autonomy support from the coach [16,25], the basic psychological needs, autonomy, competence, and relatedness [6,14,20,26], and the psychological motivational climate [8,27] can explain task and ego involvement in a group of young, elite ice hockey players. We consider both direct and indirect effects on the factors’ involvement. We have not found any studies that previously examined this relationship.

Specifically, and in line with previous research [6,11,12,14,16,20,24,27,28,29,30], the following were hypothesized:

**Hypothesis 1:** 
*Autonomy support from the coach can explain the players’ motivational involvement.*


**Hypothesis 2:** 
*All three basic needs can explain motivational involvement.*


**Hypothesis 3:** *Motivational involvement will be explained by the motivational climate*.

## 2. Materials and Methods

### 2.1. Participants

The participants were 175 male ice hockey players from Norway, ranging in age from 15 to 18 years (M = 15.91, SD = 0.45). They participated in the Norwegian Ice Hockey Federation’s elite camp. There were 7 goal keepers, 54 defenders, 99 forwards, and 15 players that were both defenders and forwards. A total of 51 players had represented the national U-18 team, and 118 (70%) players were born from January to July. Only 20 (11%) were in the last quartile. The inclusion criteria were that all participants must be in the Ice Hockey Federation’s elite camp. We obtained parental consent for the players’ participation in the study. The study was approved by the Norwegian Centre for Research Data.

### 2.2. Procedure

The data were collected during the Ice Hockey Federation’s elite camp (5 days). All of the participants participated. The players had one hour to complete the questionnaire. The players were divided into groups of 15. They were gathered in a room where the researcher was present to clarify any ambiguities in the questionnaire. The questionnaire was provided in a paper format, and the results were manually coded.

### 2.3. Instruments

All scales were validated in English. They were translated into Norwegian and back into English. They were also tested on a group of students of the same age.

#### 2.3.1. Autonomy Support

I used the short version of the sport climate questionnaire (SCQ) to examine the perceived autonomy support from the coach [25]. The scale consisted of six items worded in terms of “my coach” (e.g., “I feel understood by my coach”) (α = 0.84). The scale has been used before in the sports domain, and evidence of its adequate reliability and validity was obtained [5,20]. The questions were answered on a Likert-type scale ranging from 1 to 7 (where 1 = does not agree at all, and 7 = completely agrees). High average scores represented a high level of perceived autonomy support.

#### 2.3.2. Perceived Motivational Climate

The perceived motivational climate in sport questionnaire-2 (PMCSQ-2) was created to determine the athletes’ perceptions of goals when operating in an athletic setting [26,31,32]. The PMCSQ-2 has six subscales which were transformed into two higher-order scales. One was labeled an “ego-climate” (16 items, α = 0.91), which included punishment for mistakes, unequal recognition, and intrateam rivalry [33]. The second one was the “task-climate” (17 items, α = 0.90), which included cooperative learning, important role, and effort and improvement. All alpha values were satisfactory [34]. To complete the PMCSQ-2, the players were asked to consider their participation in ice hockey and to indicate, using a five-point Likert-type scale (1 = strongly disagrees; 5 = strongly agrees), whether they agreed with claims reflecting a task-involving (e.g., “in my team, players are encouraged to work on weaknesses”) or ego-involving (e.g., “in my team, players are encouraged to outdo their teammates”) climate. The scale was validated [26,32,33].

#### 2.3.3. Achievement Goal Orientation

We used the task and ego orientation in sport questionnaire (TEOSQ) to measure dispositional goal orientation [35,36]. The TEOSQ has a two-factor structure representing ego involvement (six items, α = 0.86) and task involvement (seven items, α = 0.77). The players were encouraged to think about how successful they felt in ice hockey in relation to their teammates and then to indicate on a five-point Likert-type scale (where 1 = strongly disagrees and 5 = strongly agrees) whether they agreed or disagreed with the items reflecting a task orientation (e.g., “I feel successful when I work really hard”) or an ego orientation (e.g., “I feel successful when others can’t do as well as I can”). Documentation for sufficient reliability and validity was obtained [37].

#### 2.3.4. Basic Psychological Needs

Using the basic psychological needs in exercise scale (BPNES) with 12 items, I measured the satisfaction of the athletes’ basic psychological needs (BPNs) [38,39]. To complete the BPNES, the players were asked to consider the assertions on a 7-point Likert scale. The players reported their satisfaction regarding the need for autonomy (four items, e.g., “the way I exercise is in agreement with my choices and interests”, α = 0.74), competence (four items, e.g., “I feel I perform successfully the activities of my exercise program”, α = 0.70), and relatedness (four items, e.g., “my relationships with my teammates are close”, α = 0.78). Likert scale was established such that 1 = does not agree at all and 7 = completely agrees. The questions were adapted for ice hockey training. The scale was validated in several studies [40,41].

### 2.4. Data Analysis

All statistical analyses were conducted using IBM SPSS (version 28.0, IBM, Armonk, NY, USA). The validity of the scales was analyzed via a confirmatory factor analysis (CFA), and Cronbach’s alpha was employed to assess the internal reliability of each scale [42]. Descriptive statistics and bivariate correlations were calculated (Table 1). 

I carried out a multiple regression analysis in which “ego–task involvement” was the dependent variable. The independent variables included autonomy support from the coach, the need for autonomy, the need for competence, the need for relatedness, the task climate, and the ego climate. The Durbin–Watson test was used to detect the presence of an autocorrelation in the residuals, and the variance inflation factor (VIF) was used to measure the impact of collinearity among the variables. Both were satisfied. Homoscedasticity was tested using the White test.

## 3. Results

### 3.1. Descriptive Statistics and Correlations

The ego and task involvement were computed into new variables with only two values: low and high ego/task (Table 2).

Values ≤ 3 were considered low and values >3 were considered high [43,44]. We computed one new variable out of these two variables. This variable, “ego–task involvement”, was categorized as follows: low ego–low task, low ego–high task, high ego–low task, and high ego–high task [43,44]. Paired-sample *t*-tests were conducted to test the differences in the means between the variables.

The descriptive statistics for all variables are presented in Table 1. The Cronbach internal reliability coefficients for all scales were satisfactory (alpha range = 0.70–0.91). The players had significantly (sign = 0.01) higher scores for task involvement (4.58) than ego involvement (3.34).

The “perceived autonomy from the coach” had a score of 4.89, which was only slightly above the median. The players had a high score for all three of the basic psychological needs, with the highest score for the “need for relatedness” (6.20). The task climate (4.10) had a significantly higher score (sign = 0.01) than the ego climate (2.90).

There were positive correlations between all three basic needs, task involvement, and task climate at the 1% level. Ego involvement had a positive relationship with the ego climate (sign = 0.01) and ego–task involvement (sign = 0.01) and a negative relationship with the task climate (sign = 0.01) and the need for relatedness (sign = 0.05).

There were significant differences (sign = 0.01) between ego and task involvement; between the ego and the task climates; among the need for autonomy, the need for competence, and the need for relatedness; and between the need for competence and the need for relatedness.

None of the players were classified as being low on task involvement. A total of 69 players (42.6%) were low for ego and high for task, and 93 (57.4%) were high for both ego and task involvement (Table 1).

### 3.2. Regression Analysis

A multiple regression analysis was conducted (Table 3) to predict ego–task involvement using autonomy support from the coach, the need for autonomy, the need for competence, the need for relatedness, the task climate, and the ego climate. The only two variables that statistically significantly predicted the ego–task involvement were autonomy support from the coach (std. beta = 0.28; sign = 0.05) and the ego climate (std. beta = 0.34; sign = 0.01). The model is significant (*F* = 3.82; sign = 0.01, *R*^2^ = 0.11) (Table 3).

## 4. Discussion

This study investigated the relationship among the perceived autonomy support from the coach, basic psychological needs, the motivational climate, and achievement goals in 15–18-year-old Norwegian elite ice hockey players. As far as we know, this is the first study to examine the concurrent effects of perceived autonomy support, the three basic psychological needs, and the motivational climate on an athlete’s task and ego involvement (achievement goals). We performed a regression analysis in this study, and the regression coefficient was low (*R*^2^ = 0.11). This means that there must be other independent variables that would explain the dependent variable. There were positive correlations between all three basic needs, task involvement, and the task climate at the 1% level. Ego involvement had a positive relationship with the ego climate and ego–task involvement and a negative relationship with the task climate and the need for relatedness. These results are supported by earlier research.

According to the descriptive analyses, the ice hockey players showed a medium level of perceived autonomy support from their coach, and they had high scores for task climate and task involvement. Contrary to this, they had mediums score for ego climate and ego involvement. This is supported by earlier research [45,46]. A total of 57% of the players had high scores for both task and ego involvement, and 43% had a high score for task involvement and a low score for ego involvement. This is supported by earlier studies [18].

There is a strong relationship between all three basic psychological needs and the perceived autonomy support from the coach and (sign = 0.01). An earlier study on young Spanish footballers found a positive relationship between autonomy support from the coach and an index of the satisfaction of psychological needs [7]. We also know that through studies among youth swimmers, it was uncovered that autonomous behavior from the coach positively predicted the satisfaction of the athlete’s needs for competence and relatedness [45]. Autonomy support is also strongly negatively associated with distress and positively associated with an athlete’s well-being [9]. Studies have also shown a strong positive association between an athlete’s basic psychological needs for autonomy, competence, and relatedness and autonomy support from the coach [7,9]. There is a very strong association between different climate or behavioral supports for an athlete’s basic psychological needs like competence support, relatedness support, structure, involvement, and a task involving climate [9].

Our first hypothesis was that autonomy support from the coach has an impact on motivational involvement. There was a positive relationship between the motivational involvement and autonomy support from the coach. This means that those players who scored high for both ego involvement and task involvement had high scores for autonomy support from the coach. According to the theory, we would expect that a high score for autonomy support from the coach would have a negative relationship with a high score for ego–task involvement [9,11]. The fact that the players were high for both task and ego involvement can explain this positive relationship with autonomy support, even if it is a bit surprising. Hence, the first hypothesis was rejected.

The next hypothesis was that all three basic needs will have an impact on motivational involvement. In our analysis, none of the basic needs had any impact on motivational involvement. All three had a positive correlation with task involvement. On the other hand, there was no significant correlation between the need for autonomy or the need for competence and the task–ego involvement. Instead, there was a negative correlation (sign = 0.05) with the need for relatedness. We know that most players scored high for both ego and task involvement, and this could have led to the fact that the basic needs did not have any impact on motivational involvement. This is not in line with the results of earlier research, which indicated that all three needs are essential for human motivation and if any are missing, there will be distinct functional costs [14,15,16,17]. An explanation for this result could be the high score for both ego and task involvement. Thus, this hypothesis was also rejected.

Our last hypothesis was that motivational involvement will be explained by the motivational climate. The ego climate had a positive impact (sign = 0.01) on the task–ego involvement. This hypothesis was confirmed. The ego climate will cause more players to score highly for ego orientation, even if most of the players had higher scores for task involvement than for ego involvement. We know that a task-oriented climate has a perceived advantage over an ego-oriented climate [12,18]. In an ego-oriented environment, athletes tend to use social comparison processes when assessing their own competence [12]. It is important to cultivate a task-oriented climate where athletes value skill mastery, intrinsic motivation, and effort rather than an ego-oriented climate in which the paramount goal is to defeat others. It might be that the coaches for these athletes encourage both a task- and ego-oriented climate, depending on the situation. The focus from the coach on daily training and competition can be different. Coaches, even if they say that they are task-oriented, can become more ego-oriented depending on the importance of the match, and this will probably also have an influence on the players’ perception of the team climate [16]. We also know that perceived ego-oriented climates are associated with negative feelings and effects [24]. One possible reason for this could be that athletes are encouraged to focus on factors out of their control in an ego environment.

Competition can improve the quality of performance and lead athletes to consistently perform their best. Competition between two players for a spot on a hockey team should lead to a better team because each player hopefully will contribute to team success and try to improve as much as she/he can in order to make the team better. Coaches must also learn to understand that in pressured situations, the quality of performance can be reduced. They must counteract this so that their athletes do not adopt negative solutions to reach a certain competition standard [47].

## 5. Conclusions

The athletes had higher scores for task rather than ego involvement, but when we transformed the data into two variables (high and low) for task and ego involvement, we found that most players scored highly for both. We also found that autonomy from the coach had a positive relationship with players that had high scores for both task and ego involvement. This was a bit surprising.

Furthermore, we found that the three basic psychological needs had no impact on the motivational involvement of the athletes.

Lastly, we found that the ego climate had an impact on motivational involvement. There was a positive relationship between a high score for the ego climate and a high score for both ego and task involvement.

### Limitations

There were some limitations in this study. If we had decided that low should be less than three and high should be equal to three or more, the results of the new combined variable, ego and task involvement, would have been different, and this could have affected the analysis.

The fact that the questionnaire was provided in a paper format and not digitally meant that it was possible to skip questions and tick several answer options, which meant that some answers had to be deleted from the survey. In addition, there will be sources of error in connection with entering the answers into SPSS.

This survey was also conducted at the same camp the year before. Some of the players participated in both camps and had thus filled in the questionnaire once before. This may also have influenced their responses.

## Figures and Tables

**Table 1 ijerph-20-06977-t001:** Descriptive statistics and internal consistency for each measure and bivariate correlations among study variables.

	1	2	3	4	5	6	7	8	9
1. Task climate									
2. Ego climate	−0.47 **								
3. Need for autonomy	0.61 **	−0.20 *							
4. Need for competence	0.56 **	−0.24 **	0.72 **						
5. Need for relatedness	0.60 **	−0.30 **	0.54 **	0.64 **					
6. Autonomy support	0.60 **	−0.39 **	0.46 **	0.42 **	0.40 **				
7. Ego–task involvement	−0.20 *	0.34 **	−0.09	−0.12	−0.20 *	0.04			
8. Ego involvement	−0.26 **	0.42 **	−0.11	−0.08	−0.18 *	0.03	0.81 **		
9. Task involvement	0.35 **	−0.14	0.31 **	0.44 **	0.36 **	0.24 **	0.03	0.02	
N	160	158	163	163	164	166	162	162	169
M	4.10	2.90	5.65	6.00	6.20	4.89	1.57	3.34	4.58
Str.d	0.60	0.82	0.89	0.69	0.77	1.10	0.50	0.91	0.42
α	0.90	0.91	0.74	0.70	0.78	0.84	-	0.86	0.77

** Correlation is significant at the 0.01 level (two-tailed). * Correlation is significant at the 0.05 level (2-tailed).

**Table 2 ijerph-20-06977-t002:** Cross-tabulation of task and ego involvement among the players, divided into low and high values.

		Task Involvement	(α = 0.77)	Total
		**Low**	**High**	
Ego involvement (α = 0.86)	Low	0	69	69
	% of total	0	42.6%	42.6%
	high	0	93	93
	% of total	0	57.4%	57.4%
Total		0	162	162
% of total		0	100%	100%

**Table 3 ijerph-20-06977-t003:** Hierarchical regression analysis of how autonomy support, the need for autonomy, the need for relatedness, and task and ego climates predict the variable ego and task involvement.

	Model 1	Model 2	Model 3
	B	B	B
Constant	1.56 **	2.25 **	1.26 *
Autonomy support	0.00	0.10	0.28 *
Need for autonomy		0.02	0.05
Need for competence		−0.04	−0.04
Need for relatedness		−0.20	−0.10
Task climate			−0.14
Ego climate			0.34 **
*R* ^2^	−0.01	0.01	0.11
*F*	0.00	1.20	3.82 **
Δ*F*		1.61	8.75 **

** Correlation is significant at the 0.01 level (two-tailed). * Correlation is significant at the 0.05 level (2-tailed).

## Data Availability

The data presented in this study are available upon request to the corresponding author.

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
