# Peer review of "How Can Autonomy Support from a Coach, Basic Psychological Needs, and the Psychological Climate Explain Ego and Task Involvement?"

_ijerph, 2023, doi:10.3390/ijerph20216977_

Round 1

Reviewer 1 Report

Evaluation o manuscript ijerph-2478190

This manuscript evaluated the autonomy from the coach, this scientific subject is interesting, I will present below my evaluation point-by-point.

The Abstract needs numerical and statistical results. Use the results from Tables 2 and 3.

The introduction is too long, I suggest reducing unnecessary information such as that presented in the 4th and 6th paragraphs, they are redundant.

The acronym SDT appears at the beginning of the introduction, but its meaning is only shown below, please correct it.

Why specifically investigate hockey players between 15 and 18 years old? The introduction does not mention anything about it.

Author Response

numerical and statsiticl ersults are inserted in the abstract

paragraph 4 is deleted

SDT are explained in the beginning. 

Why inestigate this is explained in the beginning

Reviewer 2 Report

Dear Author

The work and quality with which this study has been carried out is appreciated, which is considered to have characteristics to be able to be published and, in this way, contribute to the line of research of the present investigation. However, previously it is necessary to resolve some questions of a methodological order, foundation and writing that I proceed to detail below:

Abstract: I suggest that you synthesize the abstract to adjust to the norm established by the journal of max 200 words. You must explicitly state the knowledge gap that led you to consider this research objective. Although the study design can be inferred from the information provided in the abstract, it must explicitly state what this study design was.

Keywords: For format, you must add the keyword Keywords. I suggest reviewing the selected keywords again. Specifically, mentioning "basic needs" I don't think is entirely correct. On the other hand, I suggest adding the abbreviation or the name of the Theory that is at the base of this study, as well as a keyword that describes the population that was studied.

Introduction: You should reconsider the wording of the introduction section of your study, mainly because it is too long and does not have a logical order. An introduction is expected to have a general-to-particular statement, starting with a general problem and then narrowing down to a specific problem related to that problem. In this sense, in the wording of his introduction it is not clear what is his general problem from which the study interest arises. Likewise, although a specific study problem is posed, it lacks support as to what is the knowledge gap that should be studied. In addition, although abundant information is provided on the theories on which this study is based and its relationship with the variables of interest, the way in which said information is delivered lacks organization. Therefore, I suggest, first of all, to synthesize the information that you want to deliver. So, the integration of subheadings would help to organize the information, in addition to guiding the reader. Based on the information provided, I suggest the following subheadings: 1) Theory of Basic Psychological Needs in Sport; 2) Theory of Performance towards the Objective, and its relationship with the Ego and the Task in sport. The name of the subtitles is just a proposal that you can modify as you see fit.

Finally, although the objectives and hypotheses are clear, it is essential to support them with a more robust state of the art, that is, with similar studies that have addressed this problem. Although some studies are cited in support of the stated hypotheses, no detailed information is provided to make them plausible. Therefore, it is necessary to explicitly present information about these studies that support the hypotheses. By providing more information about these investigations, the knowledge gap will be clarified and substantiated, which in turn will support the stated objectives and hypotheses.

Line 30-34: This information should be removed here. You can move it to the end of the introduction and complement what you have there

Line 116-119: The objective written here is not consistent with what is mentioned in the abstract. Similarly, it does not fit the statistical analysis carried out in the study. I suggest using the goal declared in the abstract.

Line 119-123: The wording of these lines must be reformulated, since they are initially presented as hypotheses; however, the study hypotheses are stated more clearly and precisely below. In these lines, a kind of state of the art can be appreciated, since there are several citations that support the assumptions declared here. Therefore, as I suggested in previous comments, these lines should be placed before the general objective stated in the introduction, providing more details about the studies cited here. These studies lead to the formulation of the stated hypotheses and also constitute their state of the art. It is important that you strengthen this aspect of your article, since it is not clear what is known about the research topic of your study and, consequently, the delimitation of what is not known or the "gap in the literature" is weakly supported. As a result, your overall objective is also weakly supported.

Lines 126-129: I suggest replacing the term "impact", since it is methodologically associated with studies that allow causality analysis, such as studies with experimental designs. In this sense, you can continue to talk about "explain", "predict" or simply "relationship". In addition, since it is a regression analysis, it is important to give direction to the relationship between the mentioned variables.

Materials and methods: The study design must be declared and if it was guided by any guidelines or guidelines for its development.

Participants: You must indicate the eligibility criteria (inclusion and exclusion) and the methods of selection of the participants, that is, the type of sampling (probabilistic or non-probabilistic), and the type of access to the sample (voluntary, by snowball, called by social networks, etc.). In this line, in case you have eliminated the participants, you must declare the initial and final sample size, as well as the reasons why the participants were eliminated. It would also be desirable to state whether any sample size calculation was performed and what that size was based on the statistical analyzes that were performed or intended to be performed.

Since there were participants under 16 years of age, you must declare that an informed consent was applied, in addition to parental consent. If this was not done, it would be a serious breach of ethical standards in scientific research. In this sense, you must also declare if you have adhered to any type of international ethical guideline, such as the Declaration of Helsinki. Finally, it is necessary to provide the code provided by the institution that resulted in the development of your study.

Procedure: It would be desirable to have more information about the evaluation application process. It would be helpful to describe the setting in which they were included, the specific locations where the assessments were conducted and the relevant dates conducted, the periods of recruitment, exposure, follow-up and data collection. In addition, it is important to know the duration of the questionnaire application process in order to have an idea of the time required to complete the evaluation. It would be relevant to know how the process of signing the parental consent was managed to ensure ethics in the research. Was the application of the questionnaire in computerized format or on paper? Including these details will provide a more complete picture of the evaluation application process and will facilitate a better understanding of the methodology used in the study.

Instruments: If it is duly validated, the original language of the instrument must be indicated for each evaluation instrument and, if it is adapted and validated for the study population, said adaptations and validations must be mentioned. In addition, it is important to provide information on the psychometric properties that have been reported for the instruments in studies conducted in similar populations (Norwegian athletes or young people), explicitly reporting the reliability and internal validity indicators of each instrument. In the statistical analysis section, it is mentioned that a confirmatory factor analysis was carried out and Cronbach's alpha coefficients were calculated. It is relevant that this information is reported for each instrument in this section.

Statistical analysis: I suggest mentioning how the descriptive analysis of the data was carried out, as well as how they were addressed in the study, if any, the missing data and outliers. The work mentions the generation of categories of low and high ego and task. However, in the instrument sections, where the self-report questionnaires for said variables are described, nothing is mentioned about the cut-off scores that make it possible to define and generate said categories. Therefore, it is necessary to base the criteria used to consider a level of ego or task as low or high with previous studies. For the regression analysis, it is essential to detail whether the statistical assumptions necessary for this parametric test were met. In particular, it should be mentioned how the linearity, homoscedasticity and normality of the data submitted to the regression analysis were analyzed.

Line 187: This information is less relevant, it must be declared but at the end of this section

Table 1: This table should be included in the Results section of your study. I suggest that this Table 1 be strengthened, and in addition to the information provided, include sociodemographic information on the sample, if it exists, that allows it to be characterized. Likewise, it would be relevant to add the values obtained in the self-report instruments applied during the study. This will allow a more complete and detailed vision of the results obtained in relation to the sample and the instruments used.

Table 2: Table 2 does not provide all the information on the descriptive analysis of the study variables, this is an error that must be corrected. The information on Cronbach's alpha does not need to be reported here, as I mentioned in previous comments, this should be included in the variables and instruments section. In addition, it is not necessary to report the correlation analysis between the variables included in the regression analysis. What is essential is that the independent variables selected for analysis have a logical or theoretical relationship with the dependent variable and with each other.

Discussion: Although the results are presented in the discussion section and discussed in light of the evidence, including points of agreement and contrasts with the existing literature, it is necessary to incorporate a section on strengths and limitations that addresses possible sources of bias or imprecision in the study. In addition, it is desirable to discuss the generalizability (external validity) of the results obtained and their applicability to other populations or contexts.
